# Vitamin D and Hypoxia: Points of Interplay in Cancer

**DOI:** 10.3390/cancers14071791

**Published:** 2022-03-31

**Authors:** Ioanna-Maria Gkotinakou, Ilias Mylonis, Andreas Tsakalof

**Affiliations:** Laboratory of Biochemistry, Faculty of Medicine, University of Thessaly, Biopolis, 41500 Larissa, Greece; iogotina@med.uth.gr

**Keywords:** vitamin D, calcitriol, hypoxia, HIF, cancer

## Abstract

**Simple Summary:**

Vitamin D, conventionally considered a nutrient, is a potent hormone regulating many physiological functions. In addition, many studies point to the anticancer activities of calcitriol. However, cancer cells use mechanisms that negate the beneficial effects of calcitriol. Many of these mechanisms control or are controlled by the Hypoxia Inducible transcription Factors (HIFs) that are overexpressed in human cancers due to the development of hypoxia inside the tumors. This review discusses the crosstalk between calcitriol and HIF signaling in order to better understand their relationship to cancer, its prevention, and treatment.

**Abstract:**

Vitamin D is a hormone that, through its action, elicits a broad spectrum of physiological responses ranging from classic to nonclassical actions such as bone morphogenesis and immune function. In parallel, many studies describe the antiproliferative, proapoptotic, antiangiogenic effects of calcitriol (the active hormonal form) that contribute to its anticancer activity. Additionally, epidemiological data signify the inverse correlation between vitamin D levels and cancer risk. On the contrary, tumors possess several adaptive mechanisms that enable them to evade the anticancer effects of calcitriol. Such maladaptive processes are often a characteristic of the cancer microenvironment, which in solid tumors is frequently hypoxic and elicits the overexpression of Hypoxia-Inducible Factors (HIFs). HIF-mediated signaling not only contributes to cancer cell survival and proliferation but also confers resistance to anticancer agents. Taking into consideration that calcitriol intertwines with signaling events elicited by the hypoxic status cells, this review examines their interplay in cellular signaling to give the opportunity to better understand their relationship in cancer development and their prospect for the treatment of cancer.

## 1. Introduction

Vitamin D is a fat-soluble secosteroid prohormone produced in the skin during exposure to sunlight’s ultraviolet B radiation (UVB, 290–320 nm) that is also obtainable from the diet. Vitamin D undergoes a two-step metabolic activation in the liver and kidney to synthesize a biologically active hormonal form named calcitriol, which binds to the vitamin D receptor (VDR) to enable its diverse physiological functions [1,2]. The archetypical role of vitamin D is to maintain calcium and phosphate homeostasis, which are essential for bone morphogenesis and remodeling. Calcitriol is also an important regulator of the immune system and exhibits antiproliferative properties when applied in different cell types [3,4,5,6]. Thus, over the past decades, extensive studies have suggested that vitamin D deficiency is associated with low sunlight exposure and increased risk of many other extra-skeletal diseases such as cancer [7,8,9,10]. Many epidemiological studies revealed an inverse correlation between serum 25-hydroxyvitamin D (25(OH)D3) levels and high risk of colon [11], breast [12], prostate [13,14], gastric, and other cancers [15]. Moreover, there is strong evidence from cell and animal-based studies to support the antitumorigenic effects of vitamin D [8,16,17]. As such, it is now becoming apparent that vitamin D deficiency can contribute to the development and progression of many types of cancer. Thus, maintaining sufficient serum vitamin D levels could be beneficial for the prevention of cancer and favorable patient outcome.

Because the numerous epidemiological and experimental data indicate the beneficial role of vitamin D in preventing and treating several cancer types, clinical use of calcitriol or its synthetic analogs (referred to as vitamin D analogs) has been investigated [17]. Hypercalcemia, the major side effect of vitamin D, has strongly hindered the calcitriol clinical applications [18,19]. Moreover, accumulating data suggest that cancer cells employ several mechanisms that either reduce cellular calcitriol levels by overexpression of calcitriol deactivating enzyme CYP24A1, remarkably induced by hypoxia, to catalyze its inactivation [20] or diminish its function to protect themselves from the antitumorigenic effects of vitamin D [16,17]. Such maladaptive mechanisms are often dysregulated in solid tumors due to the development of a microenvironment characterized by reduced oxygen availability (hypoxia) caused by irregular vascularization and increased cell proliferation rates. Hypoxia triggers an adaptive machinery that relies on stabilizing the hypoxia-inducible factors (known as HIFs) [21]. HIFs are transcriptional activators that trigger a chain of events that includes reprogramming of metabolism, angiogenesis, and erythropoiesis and ultimately promote cell proliferation, survival, invasion, and metastasis [22,23]. As such, HIFs are often correlated with resistance to conventional therapy options and negative patient prognosis, much like vitamin D deficiency. Given the essential impact of HIFs and hypoxia in cancer, it is no wonder that HIFs are meaningful targets for agents with anticancer abilities, including naturally occurring compounds [24,25,26]. Furthermore, apart from HIF induction, lack of oxygen contributes to dysregulated signaling cascades [27] and compromised catalytic activity of enzymes such as the monooxygenase family members that use the molecular oxygen as a substrate and are heavily implicated in vitamin D converting reactions [28].

Given that calcitriol is regularly found to interfere with signaling cascades connected with the adaptation of cancer cells to hypoxia, we discuss the crosstalk between calcitriol and hypoxia signaling as well as possibilities and future directions to overcome the limitations of and improve vitamin D-based cancer therapy.

## 2. Vitamin D Synthesis and Metabolism

Vitamin D exists as a prohormone that needs to be transformed into biologically active products that bind to their cognate nuclear receptors to regulate diverse physiological processes. In this section, we summarize the metabolic pathways and hormonal regulation of vitamin D metabolism (Figure 1).

### 2.1. Canonical Vitamin D Metabolic Pathway

There are two major isoforms of vitamin D, vitamin D2 (ergocalciferol) and vitamin D3 (cholecalciferol) [29,30]. Both vitamin D2/3 need exposure to sunlight’s UVB radiation to be synthesized from ergosterol and 7-dehydrocholesterol, respectively. Vitamin D (both vitamin D2 and D3, calciol) originating from diet or endogenous skin synthesis is delivered to the liver by vitamin D-binding protein (VDBP). There, vitamin D is metabolized by vitamin D 25-hydroxylase (CYP2R1 and CYP27A1) to 25(OH)D (calcidiol), which is the major circulating form of vitamin D in the serum [31,32]. 25(OH)D is further metabolized by 25(OH)D 1α-hydroxylase (CYP27B1) mainly in the proximal tubule of the kidney to 1α,25-dihydroxyvitamin D (1α,25(OH)2D, calcitriol), which is the recognized biologically active form of vitamin D (Figure 1) [31,32]. Calcitriol then enters the circulation and, after binding to VDBP, is delivered to target tissues such as the intestine, bone, and kidney, where vitamin D is known to regulate absorption, mobilization, and reabsorption, respectively, of calcium and phosphate [29]. After being produced, the levels of both calcidiol and calcitriol are tightly regulated by 25(OH)D 24-hydroxylase (CYP24A1), which is the primary vitamin D inactivating enzyme catalyzing hydroxylation at C-24 and C-23 of both calcidiol and calcitriol [31,32]. This 24-hydroxylation pathway produces the biologically inactive calcitroic acid excreted in the bile [33]. The importance of this inactivation step, mediated by CYP24A1, was highlighted in *CYP24A1* knockout mice showing impaired intramembranous bone mineralization and hypercalcemia, leading to a lethal phenotype in 50% of the mice [34,35]. However, this defect was rescued in *CYP24A1* and *VDR* double-knockout mice, which suggested that it is the increased calcitriol levels and not the absence of downstream metabolites that were responsible for the flawed phenotype [35].

### 2.2. Noncanonical Vitamin D Metabolic Pathway

Alternatively, vitamin D metabolism is mediated by CYP11A1 (known as a cytochrome P450 side-chain cleavage (P450scc) enzyme) [36]. Vitamin D serves as an alternative substrate for CYP11A1 instead of cholesterol and is sequentially hydroxylated, predominantly at C-20 or C-22, without the cleavage of the side chain producing a multitude of metabolites [37]. Overall, it is estimated that this alternative path produces more than 21 hydroxy-metabolites of vitamin D [36]. Summarily, CYP11A1 products exhibit: (i) antiproliferative, differentiating, and anti-inflammatory abilities in skin cells comparable to that of calcitriol [38,39], (ii) are involved in defense pathways against UVB-induced damage and oxidative stress, and (iii) elicit anticancer abilities in a cell-specific manner [40]. As a point of interest, these alternative metabolites and normal 1,24,25-(OH)3 vitamin D3 do not activate VDR. Thus, the calcemic effects or expression of CYP24A1 can be seen in response to calcitriol.

### 2.3. Hormonal Regulation of the Canonical Vitamin D Metabolic Pathway

As a result of its diverse function, calcitriol is tightly regulated in a negative feedback mechanism [33,41]. Calcitriol inactivation primarily involves modification by CYP24A1, which is among the most prominent targets of the calcitriol–VDR–RXR complex (Figure 1) [42]. In addition, calcitriol can also induce CYP24A1 expression by recruiting histone H4 acetyltransferases and RNA polymerase II to a site approximately 50–70 kb downstream of the human CYP24A1 gene [43]. So, calcitriol signaling levels are tightly kept in control by calcitriol-driven expression of CYP24A1.

Independently, vitamin D metabolism is regulated by two hormones, parathyroid hormone (PTH) and fibroblast growth factor-23 (FGF-23), both of which maintain the calcium and phosphate homeostasis [44]. PTH, secreted by the parathyroid gland in response to calcium levels, stimulates the expression of CYP27B1, leading to an increase in calcitriol production [45]. Although calcitriol signals its degradation via CYP24A1, PTH sustains calcitriol levels by activating the renal cAMP–PKA pathway and invoking the CYP24A1 mRNA degradation [46]. FGF-23, secreted by osteoblasts and osteocytes in response to both phosphate and calcitriol levels [42], reduces serum calcitriol levels by inhibiting the expression of CYP27B1 and simultaneously enhancing the expression of CYP24A1 in the kidney [47].

## 3. Transcriptional Regulation of Target Gene Expression by Vitamin D

### 3.1. Genomic Response to Calcitriol

In its target tissues, calcitriol induces diverse biological functions both by genomic and nongenomic routes [1,48,49]. In order to elicit transcriptional responses, calcitriol binds to cytosolic VDR (a ligand-activated transcription factor and member of the nuclear receptor family), calcitriol binding promotes VDR phosphorylation, and hetero-dimerization with the retinoid-X receptor (RXR), and, finally, the nuclear translocation of the complex [50]. The calcitriol–VDR–RXR complex associates with vitamin D response elements (VDRE) in the promoter region of its target genes and recruits transcriptional cofactors to regulate the expression of its target genes (Figure 1). This process is also facilitated by protein–protein interactions (such as with the autophagy adaptor protein p62/SQSTM1) that enable the VDR–RXR binding to VDREs by directly binding to VDR and RXR [51].

As already stated, the effect of calcitriol on gene expression is manifested via the calcitriol–VDR–RXR complex. However, the expression of vitamin D targets also depends on whether the transcription start site and the VDR-binding sites are inside accessible chromatin structures [52]. It has been noted that calcitriol has the potential to elicit changes in the accessibility of chromatin, affecting a significant number of loci in leukemia (THP-1) cells [53,54]. Furthermore, VDR participates in large protein complexes with chromatin modifiers, such as KDM6B and BRD7, to provoke epigenetic alterations in cancer cells [55,56]. Usually, primary vitamin D target genes expression is effectuated within 4 h after stimulation with 1,25(OH)2D3 [54]. However, there are genes not directly regulated by calcitriol, but their expression depends on regulatory molecules encoded by primary vitamin D targets. Transcriptome-wide investigation of calcitriol response identified both primary and secondary vitamin D target genes involved in processes such as cell cycle and epigenetic control, proliferation, apoptosis, immune regulation, and angiogenesis [53].

### 3.2. Nongenomic Response to Calcitriol

Conversely, in the nongenomic pathway, calcitriol binds to membrane-bound receptors identified as 1,25D-Membrane-Associated Rapid Response Steroid-binding protein (1,25D3-MARRS). MARRS, also known as ERp57/PDIA3, is a multifunctional protein connected with the rapid cellular response to calcitriol [57]. Interestingly enough, during oxidation stress, MARRS/PDIA3 impedes PERK (PKR-like ER kinase) activation, which was found to interfere with the hypoxia signaling [58,59]. Interaction with transmembrane receptors induces rapid responses via cell signaling pathways, including phospholipase A_2_/PKC- and calcium-mediated signaling, mitogen-activated protein kinase (MAPK) cascade through direct protein–protein interactions with intracellular signaling molecules (Figure 1) [48,60,61]. As such, it was found that MARRS can mediate calcitriol signaling independent of VDR and affect mammary gland development, osteoblast maturation, or fat accumulation [62,63,64]. Moreover, sequestration of MARRS/PDIA3 or membrane-associated VDR on caveolae invaginations of the plasma membrane significantly contributes to the nongenomic calcitriol signaling [57]. According to recent studies, in a calcitriol-dependent manner MARRS/PDIA3 and VDR, separately or in concert, associate with caveolins and efficiently activate phospholipase A_2_, c-Src, and MAPK signaling to elicit intracellular changes [65,66,67].

## 4. Vitamin D Crosstalk with HIF Signaling

Since most of the known consequences of hypoxia are elicited via the HIF family of transcription factors, in this paragraph we discuss the HIF-regulation pathway and the possible interconnection with calcitriol-elicited changes.

Hypoxia-Inducible Factors (HIFs) act as heterodimers consisting of an oxygen-labile HIF-α subunit and a constitutively expressed HIF-β subunit (also called ARNT). The active heterodimer binds to specific DNA sequences of their target genes, named hypoxia response elements (HREs), recruits general co-activators such as CBP/p300, and leads to the expression of several hundreds of genes. Three HIF-α subunits (HIF-1α, HIF-2α, HIF-3α) have been described, of which HIF-1α and HIF-2α are the most studied and considered the predominant activators of hypoxia-induced gene transcription [68,69,70,71,72].

The canonical HIF-α regulation pathway requires their oxygen-dependent hydroxylation at proline (Pro) and asparagine (Asp) residues, respectively. Under normal oxygen conditions, HIF-α subunits are hydroxylated at two conserved proline residues located in the ODDD domain by the proline hydroxylase enzymes (PHD, Prolyl, Hydroxylase Domain). This modification promotes the binding of the Von Hippel Lindau tumor suppressor protein (pVHL), driving to HIF-α ubiquitination and its subsequent, rapid degradation in the proteasome. An additional hydroxylation at a conserved asparagine residue in the C-TAD domain HIF-α by an enzyme known as the HIF inhibitor (Factor Inhibiting HIF, FIH) inhibits the recruitment of CBP/p300 to HIF-α, thereby attenuating HIFs transcriptional activity. In hypoxic conditions, however, the aforementioned hydroxylation events are inhibited due to the inactivation of PHDs and FIH, resulting in the stabilization of the HIF-α protein and activation of HIFs [73]. Although HIF-1α and HIF-2α possess a high amino acid homology and are able to bind to the same HRE sequences, they occupy distinct genomic sites and activate different sets of genes, depending on the cell types [74]. Moreover, HIF-1α and HIF-2α display divergent subnuclear localization, with HIF-2α accumulating in specific nucleus structures (speckles), close to active RNA polymerase, and HIF-1α uniformly distributed within the nucleus [75]. Additionally, there is contradicting evidence that a substitution mechanism exists when a single HIF-α isoform is suppressed. Some studies demonstrate that neither HIF-1α nor HIF-2α could replace the insufficient DNA binding due to the absence of the other HIF-α isoform. In contrast, others have shown that the silencing of a single HIF-α variant could be replenished by the overexpression of the other HIF-α isoform [76,77], thus promoting cancer cells’ survival.

Numerous data demonstrate that multiple regulatory mechanisms enable strict control of HIF-α subunits to ensure fine tuning and subsequent cellular adaptation to hypoxia. These regulatory mechanisms involve extensive post-translational modifications of the alpha subunits [78] and their association with various proteins.

### 4.1. Transcriptional Regulation of HIF-α and Calcitriol

One of the pathways that result in the increased production of *HIF1A* mRNA involves activation of the JAK/STAT3 pathway [79,80,81,82,83]. Furthermore, PI3K/AKT and ERK1/2 signaling affect HIF-1α transcription in response to reactive oxygen species (ROS) generated by arsenite [84]. In particular, ROS enable the recruitment of nuclear factor erythroid 2-related factor 2 (NRF2) to an antioxidant response element (ARE) located approximately upstream of the HIF-1α transcriptional start [84,85] (Figure 2).

As already mentioned, inflammatory conditions affect HIF-1α mRNA levels by the upregulation of the NF-κB transcription factor signaling pathway [86,87,88]. More specifically, phosphorylation of the IkB protein (Inhibitory kB) and subsequent activation of NF-κB subunits (p50 and p65 subunits (RelA) affects *HIF1A* mRNA in response to thrombin, H_2_O_2,_ and short-term hypoxia [88,89]. Moreover, TNFα (Tumor Necrosis Factor-α) induces the transcription of *HIF1A* mRNA and protein but inhibits the hypoxic stimulation of HIF-1 transcriptional activity in airway smooth muscle cells [90] (Figure 2).

Another study has demonstrated a mechanism for maintaining nitric oxide (NO) homeostasis in macrophages in which HIF-1α and HIF-2α isoforms act competitively. When the interferon-γ concentration is low, the transcription of *EPAS1* is induced, leading to an increase in the expression of Arginase 1 and a decrease in NO production. In contrast, the elevation of interferon-γ concentration leads to the induction of *HIF-1A* and a subsequent increase in the expression of inducible Nitric Oxide Synthase (iNOS) and NO production [91].

Epigenetic modifications also control HIF-2α transcription. MBD3 (Methylated CpG binding protein 3) binds to the *EPAS1* gene promoter and facilitates its transcription [92]. In addition, HIF-2α transcription has been reported to be controlled by the IGF1R signaling (Insulin-like Growth Factor 1 Receptor) and the activity of PI3K (phosphoinositide 3-kinase) and the mTORC2 complex (mTOR Complex 2) [93]. In summary, IGF-II drives *EPAS1* mRNA expression in hypoxic neuroblastoma cells, which is executed via IGF1R/INSR–PI3K–mTORC2 signaling, whereas HIF-1α is regulated only at the protein level via PI3K–mTORC1. Cezanne deubiquitinase is also involved in the regulation of HIF-2α mRNA synthesis (Figure 3). Through its deubiquitinase activity, Cezanne controls the levels of the E2F1 transcription factor protein (E2F transcription factor 1) that directly binds to the *EPAS1* gene promoter [94]. Moreover, *EPAS1* gene expression is downregulated by the deacetylases of HDAC class I/II histones. Decreased expression of HIF-2α enhances calcium signaling, leading to increased mTORC1 complex activity and induction of cell proliferation in sarcoma mice [95].

Given the oncogenic potential of activated STAT3, the ability of calcitriol to repress signaling mediated by this transcription factor is paramount among its antineoplastic effects. In this regard, the constitutive activation of STAT3 has been shown to mediate growth, survival, and invasion of breast cancer cells [96]. At the same time, vitamin D analogs, such as Gemini, could markedly repress CD44-STAT3 signaling, suggesting its potential to inhibit breast cancer invasion [97]. Moreover, in an in vivo and in vitro preclinical study of gastric cancer, another noncalcemic analog of vitamin D, paricalcitol, showed a robust capacity to disrupt inflammation-dependent tumor promotion. Indeed, paricalcitol significantly suppressed the expression of inflammatory mediators such as COX-2 while strongly reducing the levels of phosphorylated STAT3 by limiting the level of NF-κB in the nucleus [98].

Supporting the opposing functions between HIF-signaling and calcitriol in cancer, calcitriol has been shown to intrinsically block NF-κB activity and downregulate NF-κB protein levels in a variety of cell types [99,100,101,102]. A partial mechanistic rationale for the anti-inflammatory effects of calcitriol comprises the stimulation/stabilization of the NF-κB inhibitory protein α (IκBα), the physical interaction of the VDR with IκB kinase β protein (IKKβ), and the blocking of NF-κB binding to DNA, all of which result in NF-κB inhibition [103,104]. These inhibitory effects of calcitriol upon NF-κB are highly relevant for medical oncology, given the critical role that NF-κB plays in cancer pathogenesis and that it is constitutively expressed in several types of malignant tumors [105]. Indeed, NF-κB activation has been shown to regulate the expression of many genes involved in oxidative stress, cellular transformation, proliferation, inflammation, antiapoptosis, angiogenesis, invasion, metastasis, and numerous other potentially carcinogenic processes [105,106,107].

One study has demonstrated that calcitriol-mediated antiproliferative effects on tumor-derived endothelial cells (TDEC) are VDR dependent and that loss of VDR in knockout models can lead to an increase in HIF-1α, VEGF, Ang1, and PDGF-BB levels and subsequent abnormal tumor angiogenesis [108]. These results corroborate our observations that silencing of VDR in the absence of calcitriol induces the transcriptional activity of both HIF-1 and HIF-2. This could imply the existence of an additional regulatory mechanism of HIF-1/2 by VDR alone that does not require calcitriol binding to VDR. One speculation could be that VDR, in the absence of calcitriol, is bound either to the promoters of *HIF1A/EPAS1* genes or the promoters of HIF-1/2 target genes. Moreover, VDR leads to the deactivation of NF-κB by creating a complex with the IKKβ protein. At the same time, NF-κB transcriptionally activates HIF-1, as has already been discussed [87,88,103]. Thus, these mechanisms explain the amplification of HIF transcriptional activity after VDR silencing (in the absence of calcitriol).

In other studies, calcitriol inhibits ROS-NLRP3-IL-1β signaling axis via activation of Nrf2-antioxidant signaling in hyperosmotic stress stimulated human corneal epithelial cells [109]. In many malignant cells, calcitriol modulates growth-factor actions such as upregulation of the expression of the insulin-like growth factor binding protein-3 (IGFBP-3) gene in PCa cells, which in turn leads to an increase in the expression of p21, causing cell cycle arrest and inhibits NF-κB activation and indirectly influencing the expression of HIF-α mRNA [88,103].

Taken together, there are substantial data indicating that calcitriol interferes with important signaling cascades that affect HIF-α mRNA expression.

### 4.2. HIF-α Translation and Calcitriol

Elevated HIF-1/2α mRNA translation levels increase protein levels and, as expected, HIF-1/2α activity, mainly in cells exhibiting activation of PI3K-AKT-mTOR pathway, which is a common feature of cancer cells. More specifically, growth factors activate a tyrosine kinase receptor, which in turn activates PI3K and MAPK. PI3K activates the serine/threonine kinase AKT (also known as protein kinase B, PKB) and the mTOR protein (mammalian Target of Rapamycin). In the MARK pathway, ERK1/2, which has been activated by the MEK, in turn, activates the MNK (MAPK interacting protein kinase). ERK and mTOR phosphorylate the p70 S6 kinase (S6K) protein, which then phosphorylates the S6 ribosomal protein and binds to the eukaryotic translation initiation factor 4E (eIF-4E) protein (4E-BP). The binding of 4E-BP1 to eIF-4E is inhibited by its phosphorylation by mTOR and ERK and results in the translation of 5′ envelope mRNAs. MNK also phosphorylates eIF-4E and stimulates its action directly. The result of this pathway is the increased translation of a specific group of mRNAs, including HIF-1/2α [110]. The above mechanism of action of the PI3K pathway may lead to the induction of HIF-1α by hormones such as angiotensin II, thrombin, insulin, and endothelin in vascular smooth muscle cells and by lipopolysaccharides (LPS) in macro (LPS). Additionally, *HIF1A* mRNA translation rates are also dependent on PERK activation, which phosphorylates and inactivates eukaryotic initiation factor 2α (eIF2α), limiting HIF-1α synthesis [59]. Interestingly, calcitriol receptor MARRS/PDIA3 has been shown to prevent PERK kinase activation in the unfolded protein response pathway [58]. References to HIF-2α are limited, showing that IGF-1 has been described to induce HIF-2α via the PI3K pathway, thereby inducing VEGF expression in osteoblast cells [110,111].

Studies on mTOR have shown that there are two distinct mTOR complexes called mTORC1 and mTORC2. The mTORC1 complex is sensitive to rapamycin inhibition and consists of mTOR and Raptor protein (regulatory-associated protein of mTOR). mTORC1 is activated by AKT and then induces protein synthesis by phosphorylation of p70 S6 kinase and 4E-BP1. In contrast, mTORC2 is not inhibited by rapamycin, is composed of mTOR and Rictor protein (a rapamycin-insensitive companion of mTOR), and activates AKT via phosphorylation [112]. Until recently, no link was found between HIF-2α and mTOR. However, a recent study showed that HIF-2α expression depends on mTORC2, which is regulated by cellular redox status, whereas HIF-1α expression depends on both mTOR complexes in renal cancer cells. Additionally, the same work studied the dependence of HIF-1/2α expression on AKT. Out of the three different AKT isoforms (AKT1, AKT2, AKT3), it was observed that the expression of HIF-2α depends on AKT2 while the expression of HIF-1α on AKT3 [113,114].

In this HIF-1/2α regulatory pathway, calcitriol can indirectly influence HIF-1/2α synthesis by moderating PI3K activation by increasing the expression of PTEN in a VDR dependent manner [63,115].

Notably, we have recently reported a VDR-independent mechanism by which calcitriol influences both HIF-α mRNA expression levels. Treating hepatoma cells with calcitriol resulted in *HIF1A* and *EPAS1* mRNAs localization in ribosomal fractions that are associated with low translation rates. Furthermore, calcitriol treatment resulted in decreased phosphorylation levels of AKT and downstream translation initiation factors (Figure 3) [116].

Another way that calcitriol effects AKT/mTOR pathway is by inducing the expression of DDIT4 (DNA damage-inducible transcript 4 or REDD1), which enables the assembly/activation of TSC1/2, keeping mTOR and its downstream targets inactive [117,118]. Concurrently, HIF-1 induces the expression of DDIT4/REDD1, which in a negative feedback loop impairs mTORC1, HIF-1α accumulation and suppresses tumor growth [119].

### 4.3. HIF-α Post-Translational Phosphorylation and Calcitriol

Extensive investigation during the past decades has shown that HIF-1α and HIF-2α are significantly controlled by diverse intracellular signaling pathways not directly affected by oxygen levels. Most of these pathways culminate in HIF-1α/HIF-2α phosphorylation, which regulates their stability, localization, protein interactions, and subsequently activity (Figure 4) [25,78].

HIF-1α can be directly phosphorylated by several kinases, including GSK3, PLK3, ATM, PKA, CDKs, which affect its stability. GSK-3β-mediated phosphorylation targets HIF-1α on multiple residues and decreases its protein levels in a process regulated by USP28/SENP1 [120,121,122]. It has been shown that calcitriol reduces the secretion of IL-1β, which inhibits GSK-3β [123]. Another HIF-1α destabilizing modification is mediated by direct HIF-1α phosphorylation at two sites (S576 and S657) by Polo Like Kinase 3 (PLK3) that targets HIF-1α for degradation in a VHL-independent manner [124]. On the contrary stabilizing phosphorylation is mediated by ATM, PKA, CDKs [78,125,126,127]. Interestingly, calcitriol has been found to interfere with CDK-mediated signaling in cancer cells (see also Section 5) [128,129].

A direct phosphorylation on the C-TAD domain that positively affects HIF-2α activity is mediated by casein kinase 2 (CK2). It reduces HIF-2α affinity for Factor Inhibiting HIF (FIH), explaining the increased transcriptional activity of HIF-2 [130,131]. In line with this, protein kinase CK2 positively regulates the calcitriol-inactivating enzyme CYP24A1 reducing its antitumor effect [132].

Casein kinase 1δ (CK1δ) modifies HIF-2α on two amino acid residues (S383, T528), contributes to the accumulation of HIF-2α in the nucleus, and activates HIF-2-mediated transcription [133]. In opposition, CK1δ decreases HIF-1 transcriptional activity by phosphorylating the HIF-1α subunit on Ser247 inhibiting its dimerization with ARNT [134,135]. HIF-1α and HIF-2α show high amino acid homology in the PAS-B domain, where the two HIF-1/2α subunits are modified by CK1δ. Thus, the fact that CK1δ phosphorylates the two HIF-1/2α subunits into discrete residues and regulates their action in the opposite way is of great interest.

Previous work from our laboratory has established that direct phosphorylation of HIF-1α/HIF-2α by ERK1/2 has a profound effect on HIF-1/HIF-2 activity and cancer cell adaptation to hypoxia [136,137,138]. More specifically, phosphorylation of HIF-1α by ERK1/2 at residues Ser641/643 that reside inside a small domain termed ETD (ERK Targeted Domain; amino acids 616–658) masks a nearby CRM1-dependent nuclear export signal (NES), inhibits HIF-1α nuclear export and increases HIF-1 transcriptional activity [138,139]. Moreover, this phosphorylation mediates the association of HIF-1α with NPM1, augmenting HIF-1 activity and cellular response to hypoxia [137]. ERK1/2 also modifies HIF-2α and controls its nucleocytoplasmic shuttling. ERK1/2 phosphorylate HIF-2α at residue S672 and stimulates the transcriptional activity of HIF-2 by inhibiting its CRM1-dependent nuclear export [136,140]. Furthermore, in a recent report, it was shown that ERK1/2 modify HIF-2α inside the oxygen-dependent degradation domain and enhance its interaction with hypoxically induced USP33, which deubiquitinates HIF-2α and leads to its stabilization preferentially in glioma stem cells [141]. It has been shown that ERK1 and ERK2 kinases are activated in VDR-positive and -negative breast cancer cell lines. VDR+ cells show a biphasic activation, a rapid response pathway, and a VDR-dependent response. In VDR– cells, ERK activation only occurs early on [142]. Although it seems contradictory that calcitriol activates ERKs that stimulate HIF activity, the profound effect of calcitriol on HIF-1/2α mRNA translation rate possibly explains calcitriol’s anticancer properties (See Section 4.2).

## 5. Cancer–Hypoxia–Vitamin D

Cells in solid tumors frequently encounter a drop in oxygen levels that depends on cellular proliferation rates, aberrant vasculature, and distance from oxygenated perivascular areas [143]. This developing hypoxic microenvironment is a critical factor that determines the biological behavior of cancer cells and their fate. Under these conditions, cells overexpress HIFs, which activate the transcription of hundreds of target genes and drive cancer progression. Thus, HIF-1α expression in many common human cancers is associated with increased patient mortality [144]. Apart from reduced oxygen levels, HIF expression in tumors is a result of genetic alterations such as the loss or mutation of a tumor suppressor (e.g., VHL in ccRCC) [145].

Activation of HIFs drives cancer progression by initiating a cascade of events that include the metabolic reprogramming of cancer cells that, along with resistance to apoptosis, sustain cell survival and proliferation, angiogenesis, inflammation, and immune evasion and epithelial to mesenchymal transition, extracellular matrix remodeling that support invasion and metastasis [25,143]. It is very intriguing that calcitriol-mediated signaling opposes and inhibits all these cancer-promoting functions of HIFs and supports a proapoptotic phenotype (Figure 5 and Table 1).

A distinguishing characteristic of cancer cells is the upregulation of glycolysis to sustain energy and generate biomass. This switch to anaerobic metabolism is greatly enhanced under hypoxia and mediated by HIF activation and subsequent expression of glycolysis-supporting proteins such as GLUT1, HK2, and LDHA [146]. Interestingly, calcitriol significantly impaired the expression of these critical glycolytic proteins in an HT29 subcutaneous xenograft mouse model [147].

However, apart from hypoxia, nonhypoxic stimuli such as growth factors (e.g., PDGF, TGF-β, IGF-1, and EGF) and cytokines activate signaling pathways (PI3K/AKT, MAPK) that are implicated in cancer cell proliferation and growth, and also affect HIF activation [93,168,169,170,171,172]. Moreover, it has been observed that in endothelial and hepatocarcinoma cells, hypoxia can activate ERK-mediated signaling and stimulate their proliferation [173,174]. On the contrary, a number of growth factor-stimulated pathways have been shown to be inhibited by calcitriol [16,175].

Activation of Wnt-based signaling leading to the release of β-catenin from APC (Adenomatous Polyposis Coli) complex is frequently found to be dysregulated in cancers as it activates transcription of genes heavily involved in cellular proliferation [176]. Cell-based studies suggested that calcitriol induces binding of VDR to β-catenin and reduction of β-catenin-mediated gene transcription [148,149]. These observations were also supported by findings in an APC minus animal model, in which injected calcitriol reduced polyp number and expression of β-catenin target genes in the small intestine and colon [150]. Independent of VDR association to β-catenin, calcitriol was reported to restrict Wnt-activator DKK4 and increase Wnt-antagonist DKK-1 expression [151,152].

Another example of vitamin D’s antiproliferative activity is its implication with mitogenic signaling. EGFR signaling is hindered as the activated receptor is targeted by calcitriol to early endosomes [153]. Furthermore, the VDR can bind on a VDRE present in the EGFR promoter after stimulation by calcitriol [154]. Additionally, calcitriol reduces the expression and activity of the *EGFR* gene and also represses the *SPRY2* gene (Sprouty RTK signaling antagonist 2), which encodes an activator of intracellular EGF signaling [155,177]. However, the duration of mitogenic signaling is found to elicit a different cellular response to calcitriol; transient activation of MAPK signaling by EGF enhanced calcitriol-mediated gene transcription [178] while constitutive activation of MAPK signaling impairs it [179]. Additionally, vitamin D and its analogs inhibit insulin-like growth factor 1 (IGF1)-stimulated signaling. This effect was associated with increased release of IGF binding protein 3 (IGFBP3), which is known to limit the ability of IGF1 and IGF2 to interact with their respective receptors [156,157]. The *IGFBP3* gene promoter contains a VDRE, which after EMSA and ChIP analysis, is suggested to be directly regulated by calcitriol [180]. At the same time, antisense oligonucleotides against IGFBP3 abolished the calcitriol-mediated growth arrest [181]. Another critical signaling pathway involved in cell survival and growth is based on signaling on AKT activation. Calcitriol attenuated PI3K pathway activation by increasing the expression of PTEN, which negatively regulates the associated kinases. Thus, calcitriol-induced blockade of cell growth is connected to reduced AKT phosphorylation levels. It was also found that the *PTEN* promoter contains a functional VDRE guided by VDR [63,115].

Moreover, control of the cell cycle is interlaced with cellular growth. Calcitriol inhibits the proliferation of prostate cancer cells by disrupting the cell cycle in the G1/G0 [158,175] phase through a p53-dependent mechanism. In particular, calcitriol increases the expression of p21Waf/Cip1 kinase inhibitors and p27Kip1 [158,182,183], leading to reduction i cyclin-dependent kinase 2 (CDK2) activity and hyper-phosphorylation of the retinoblastoma protein (pRb). Cdk1 phosphorylation leads to cell cycle progression. Expression of *CDK1* mRNA was suppressed after calcitriol treatment while cell cycle arresting proteins such as RBL2 (Rb-like protein p130) and RBBP6 (Rb binding protein 6) were up-regulated [128,129]. At the same time, longtime treatment of cells with calcitriol or analogous agents for more than 24 h (indicative of indirect vitamin D effects) decreased the mRNA levels of cyclins A, B, and F [184,185].

Evading apoptosis is another critical aspect of solid tumors. Cancer-associated hypoxia has been involved with resistance to apoptosis by enhancing the expression of antiapoptotic proteins such as BNIP3/3L, NDRG, MCL1, and NPM1 [73]. Apart from transcriptional reprogramming, it has been shown that hypoxia triggers the proteolytic processing of mitochondrial VDAC1 and a C-terminally truncated form (VDAC1-ΔC) is produced and protects from pharmacologically induced apoptosis [186,187]. There is also evidence that a nonmodified by ERKs mitochondrial HIF-1α form protects cancer cells from apoptosis under hypoxia [188]. Concerning calcitriol, it has been demonstrated that calcitriol induces apoptosis in prostate and breast cancer cells by breaking down mitochondria and activating the endogenous apoptosis pathway. In this case, apoptosis is stimulated by suppressing the expression of antiapoptotic proteins such as Bcl-2 and Bcl-XL and by increasing the expression of proapoptotic protein Bax at the same time [159].

In a process opposing cell proliferation, calcitriol stimulates the differentiation of normal and cancer cells. Notably, calcitriol triggers a variety of immature hematopoietic myeloid cells to differentiate into mature cells, including M-1 mouse myeloid leukemia cells, human promyelocytic U9 cell cells, and human HL-60 monocyte leukemia cells. It stimulates myeloid leukemia cell lines to eventually differentiate into monocytes/macrophages [183]. In HL-60 cells, the calcitriol-induced commitment to differentiation appears to be induced via the suppression of the c-Myc oncogene expression [161]. It also seems to diminish the expression of c-myc in prostate cancer cells, including the androgen-dependent cells [162].

Additionally, there is strong evidence that inflammation contributes to the development and progression of many cancers [189,190]. Inflammatory mediators such as cytokines, chemokines, prostaglandins, and reactive oxygen species (ROS) increase oncogenesis by activating multiple signaling pathways in tumor tissues. Thus, hypoxia and HIF-1/2α stabilization are closely linked with changes in gene expression initially provoked by inflammation and therefore have a profound impact on disease progression and outcome [191]. The anti-inflammatory potential of calcitriol became evident after studies on prostate and breast cancer cells. Calcitriol exhibits its antineoplastic effects by regulating some of the major molecular pathways involved in inflammation. It has been shown that calcitriol leads to inhibition of prostaglandin synthesis (PG) [163], inhibition of stress-activated kinase signaling [164], and subsequent production of inflammatory cytokines and inhibition of nuclear factor κB (NF-κB) signaling [102,165], which directly affects *HIF1A* gene transcription [87].

Another critical step in the growth, development, and metastasis of tumors is angiogenesis, the process of forming new blood vessels from existing vessels [192]. The most prominent proangiogenic stimulant of angiogenesis is VEGF [193]. VEGF expression can be stimulated due to hypoxia, oncogenic signaling and involves ERK activation and transcriptional regulation by HIF-1α [194,195,196]. Moreover, hypoxia enhances VEGF production by stabilizing its mRNA to sustain its protein synthesis rate [197]. Thus, neosynthesized VEGF leads to the reorganization of endothelial cells and capillary formation. Furthermore, over-expression of VEGF receptors (VEGFR-2, VEGFR-3) and factors such as angiopoietin-2 that are induced in hypoxic conditions and promote aberrant vessel formation and branching associated with the aggressive cancer phenotype [198,199,200]. Contrarily, calcitriol has been reported to exert antiangiogenic effects in animal models and inhibit the proliferation of cultured endothelial cells [166,201]. Furthermore, calcitriol or its noncalcemic analog (22-oxacalcitriol) have been shown to impair neovascularization in a mouse choroidal sprouting ex vivo model [167]. Early studies have implied that this kind of activity is mediated in a HIF-1/VEGF axis [202]. Importantly, we have recently demonstrated that calcitriol reduces *VEGF* and *EPO* mRNA levels in liver cancer lines by suppressing both HIF-1α and HIF-2α protein synthesis by influencing AKT pathway activation and independently of VDR [116].

Consequently, activation of calcitriol signaling has been correlated with reduced malignancy and cancer progression. Independent genetic studies in a variety of cancer types have shown that two distinct *VDR* gene polymorphisms that either reduce VDR potency [203] or VDR mRNA stability [204] are correlated with increased risk of cancer in at least three (breast, prostate, and skin) cancer types (for comprehensive review and analysis see [205]). Furthermore, epidemiological data show that high vitamin D serum levels in colon and breast cancer patients are related to favorable outcomes [206,207,208]. Although these data indicate the great prospect of calcitriol as an anticancer agent, there are mainly two limiting factors that hinder its broader use: (i) its side effects when calcitriol is administered in high doses (oral calcitriol dose ranging from 0.5–2.5 μg/kg daily with a maximum tolerated dose being 1.5 μg or 2.5 μg depending on the study) [208], and (ii) the decreased availability of calcitriol due to its enzymatic deactivation by the calcitriol-induced CYP24A1 gene [32]. Concerning the latter, hypoxia induces CYP24A1 levels [20,209] and, at the same time, it was found that in breast cancers, CYP27B1 levels drop while CYP24A1 levels rise [210]. Moreover, tissue hypoxia (though not in cancer tissues) caused a drop in VDR, vitamin D binding protein (VDBP), and 25-hydroxylase (CYP2R1) levels while it increased CYP27B1 and CYP24A1 levels [211]. Thus, hypoxia seems to dysregulate the calcitriol-driven signaling and instead promotes its deactivation, contributing to the malignant phenotype. Hence, definitive evidence about the efficacy of calcitriol to treat cancer has not yet been established clinically. However, few clinical trials, mainly conducted in prostate cancer patients, have tried the efficacy of calcitriol to treat cancer [208,212]. In an older study performed on hormone-refractory prostate cancer patients, administration of calcitriol as monotherapy was inconclusive, as it was terminated at an early stage due to the emergence of adverse effects [213]. On the other hand, combination treatment of prostate cancer patients with calcitriol and established chemotherapeutic drugs returned mixed results [212]. Administration of calcitriol with carboplatin had a negative outcome [214], whereas the combination of calcitriol with docetaxel showed positive patient response [215]. In both cases, oral calcitriol was administered at 0.5 μg/kg the day before the intravenous administration of each chemotherapeutic (36 mg/m2 for docetaxel) for 6 weeks on an 8-week cycle [212]. In both cases, the adverse effects of combination treatment were similar to these of single-agent administration [214,215]. To summarize, clinical application of calcitriol, although promising, is hindered by its low bioavailability at the cancer site leading to administration of high calcitriol dosage and the consequent appearance of side effects [208,216,217].

Taken together, it can be suggested that HIF- and calcitriol-mediated signaling are interconnected, and that calcitriol can have beneficial effects in hypoxic cancers after taking into consideration the problems raised from clinical trials with calcitriol. The development of calcitriol analogs that show less specificity for CYP24A1 but elicit strong VDR responses could be a solution for the low availability of calcitriol at the cancer site. Parallelly, given the hypoxic status of many cancers, another opportunity for research arises from the developing field of specific HIF inhibitors [25,146] that could be tried in combination with calcitriol or its analogs in various cancer models to establish new clinical applications to treat cancer. Under this scope, a typical example of HIF-dependent cancer is clear cell Renal Cell Carcinoma (ccRCC), which occurs due to constitutive activation of HIF-α after pVHL loss of function in renal cells [218]. Notably, two prototype agents (PT2399, PT2385) designed to efficiently inhibit HIF-2 activity exhibited significant anticancer potential [219,220]. Furthermore, initial PT2385 administration to a patient suffering metastatic renal carcinoma (ccRCC), followed by a Phase I clinical trial in patients with advanced ccRCC, PT2385 showed promising efficacy and tolerability as monotherapy [220,221]. Consistently, a recent meta-analysis of nine previous studies showed an inverse correlation between circulating and dietary vitamin D levels and the occurrence of ccRCC [222], suggesting that a combination of HIF inhibitors (including natural compounds such as flavonoids, celastrol, and resveratrol [223,224,225]) and calcitriol could be a potential anticancer strategy to treat ccRCC.

## 6. Conclusions

In this review, we aimed to raise the point of the possible convergence between vitamin D impact on HIFs, their signaling, and more generally, on cancer cells adaptation, survival, and metastasis under hypoxia. The supposed mechanisms should be further confirmed, and one of the main points of our review was to indicate these objectives of future research and not simply review what has been previously achieved.

There is extensive evidence from research performed in cell lines, animals, and patient samples that signify the contribution of hypoxia and HIF activation to the development and progression of cancer. However, it is very intriguing that calcitriol elicits the opposite effects on the same cancer-promoting functions. Given that calcitriol can directly affect HIF-1/2α mRNA expression, translation, and modifications, it is highly plausible that many of its anticancer abilities are a result of calcitriol interfering with HIF-associated signaling. As hypoxia can also directly affect calcitriol hydroxylation (due to lack of oxygen) or increase the expression of its inactivating enzymes (CYP24A1; [20]), calcitriol or newly developed analogs can defend against cancer development. However, finding the optimal clinical application of the vitamin D system is a challenging and multifaceted task that demands widespread investigation, taking into consideration the hypoxic nature of many human cancers.

## Figures and Tables

**Figure 1 cancers-14-01791-f001:**
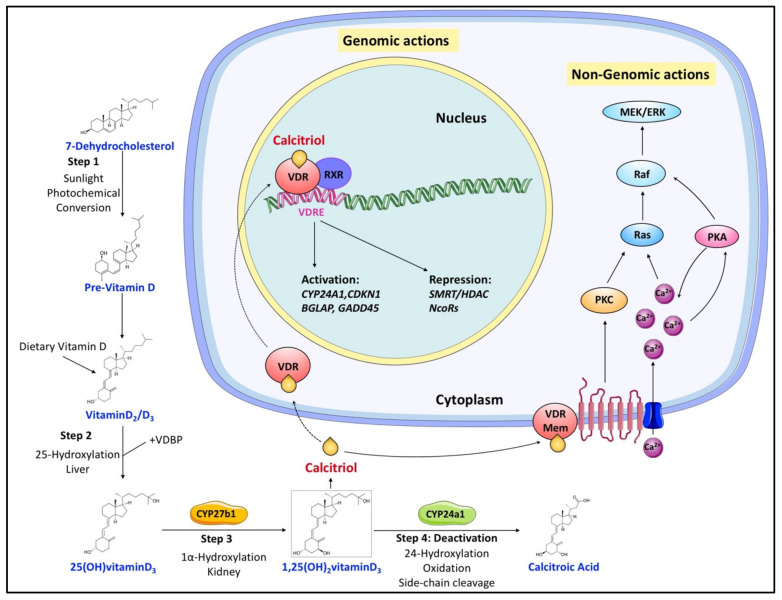
Overview of vitamin D canonical metabolism and its genomic or nongenomic effects. Dietary or cutaneously synthesized vitamin D undergoes two subsequent hydroxylation steps in the liver and kidney to produce active calcitriol (1,25(OH)_2_ vitaminD_3_). Calcitriol exerts its functions either by binding to VDR to regulate gene expression or by associating with extracellular binding sites to modulate signaling pathways that influence various cellular processes. Regulation of calcitriol levels also requires inactivation steps mainly involving its hydroxylation by CYP24a1.

**Figure 2 cancers-14-01791-f002:**
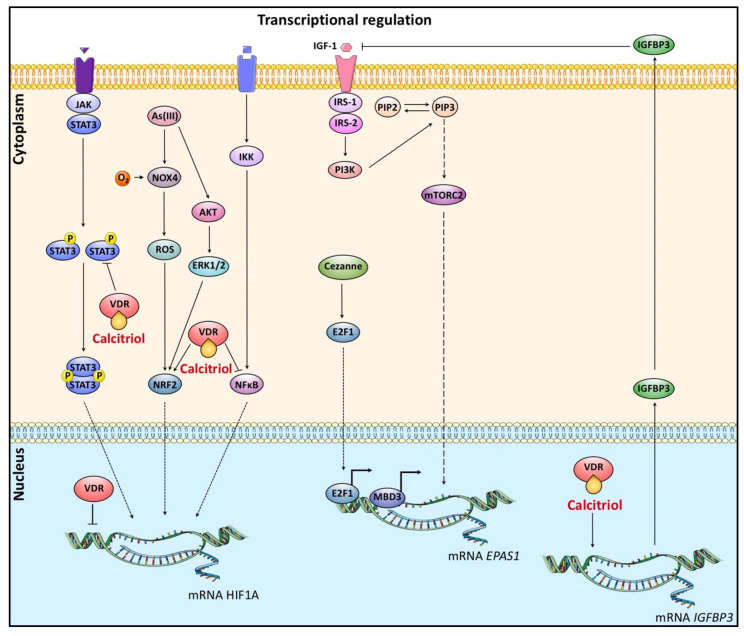
The implication of calcitriol signaling in the transcriptional regulation of *HIF1A* and *EPAS1* genes. Calcitriol binding to VDR results in the expression of proteins that control the activity of STAT3-, NRF2-, NF-κB-dependent pathways, and interferes with *HIF1A* transcription. There are also data suggesting that VDR directly inhibits *HIF1A* expression, albeit in the absence of calcitriol. Moreover, the calcitriol-VDR complex indirectly controls *EPAS1* transcription by enhancing the expression of IGFBP3 protein.

**Figure 3 cancers-14-01791-f003:**
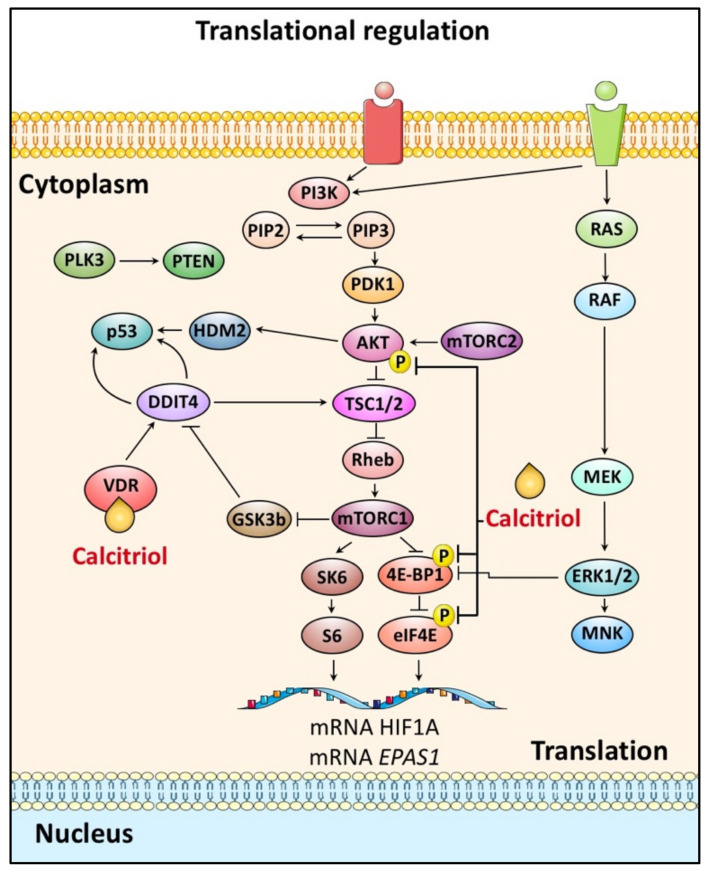
HIF-1/2α subunits regulation by calcitriol at the level of their protein synthesis. HIF-1/2α mRNA translation is mainly regulated by the PI3K/AKT pathway, which frequently cross-talks with ERKs. Calcitriol in complex with VDR enhances DDIT4 expression and impairs mTORC1-mediated translation of HIF-1/2α. Furthermore, calcitriol treatment results in decreased phosphorylation of PI3K/AKT-pathway components and limits HIF-1/2α translation rates.

**Figure 4 cancers-14-01791-f004:**
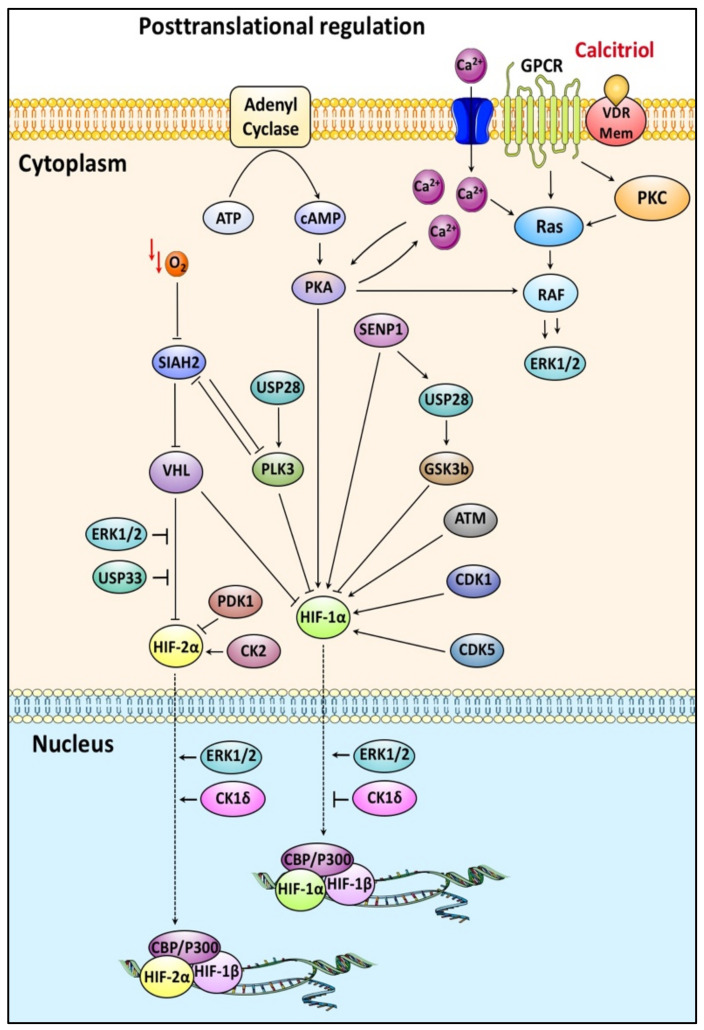
Involvement of calcitriol in the posttranslational regulation of HIF-1/2α. Both HIF-1/2α subunits are decorated by multiple phosphorylation events that control their stability, nucleocytoplasmic shuttling, and transcriptional activity. Calcitriol can indirectly interfere with these modifications by associating with plasma membrane receptors and modulating intracellular signaling cascades.

**Figure 5 cancers-14-01791-f005:**
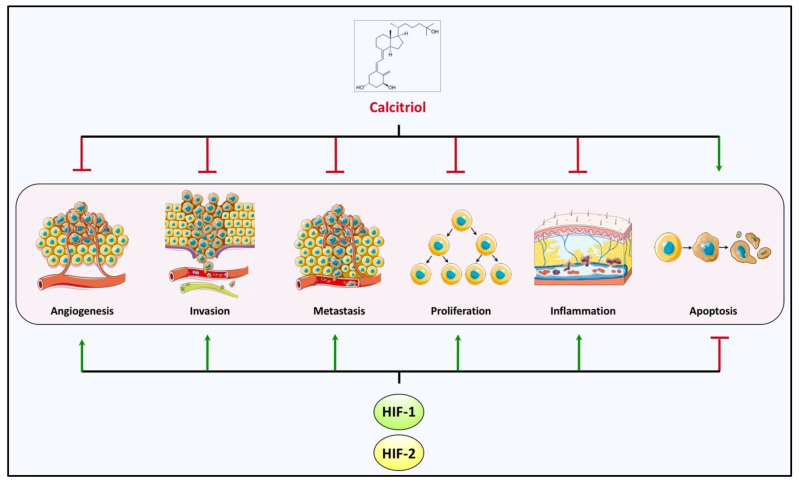
Calcitriol and HIF-1/2 cross-talking in hallmarks of cancer. HIF-1/2-mediated transcription governs essential processes implicated with adaptation to the hypoxic microenvironment of solid tumors and promotes cancer progression. There is accumulating evidence that calcitriol interferes with these particular events and opposes the advancement of the malignant phenotype, rendering calcitriol or its analogs promising agents for chemoprevention and treatment.

**Table 1 cancers-14-01791-t001:** Signaling pathways or cellular processes affected by calcitriol in various cell- or animal-based models.

Model	Pathway/Process Affected by Calcitriol	References
HT29/SW480 cell lines and HT29 NOD/SCID mouse xenografts	Suppression of glycolysis and reduced tumor xenograft volume	[147]
SW480, SW620, SKBR-3, HEK 293, NCI-H28 cell lines	Inhibition of β-catenin–TCF mediated transcription	[148,149]
C57BL/6J Apc ^+/+^ and Apc ^−/+^ mice	Reduced β-catenin signaling and polyp number	[150]
SW480-ADH cells and SW480-ADH xenografts in immunodeficient mice	Induction of Wnt antagonist DKK-1	[151]
SW480-ADH, HEK293Tcells	Reduction Wnt-activator DKK-4	[152]
A431, NR6, HeLa, BT549, Caco-2 cells	EGFR targeting to early endosomes reduction of ERK1/2 activation	[153,154,155]
MCF-7, Hs578T, prostate epithelial, and immortalized prostate epithelial P153 cells	Inhibition of proliferation due to reduced IGF signaling	[156,157]
LNCaP and DU145 cells and prostate specific PTEN-knock out mouse	Inhibition of prostate cancer cell growth	[63]
LNCaP-FGC cell line	Cell cycle arrest and decreased cell proliferation due to CDK-2 downregulation	[158]
LNCaP and Y79 cell lines	Increased apoptosis; decreased Bcl-2 and Bcl-XL and increased Bax expression	[159,160]
HL-60, LNCaP, C4-2, and RWPE-1 cell lines	Decreased c-Myc expression and cell proliferation; promoted differentiation of HL-60 cells	[161,162]
LNCaP, PC-3, MRC-5 cell lines, and prostate adenocarcinoma samples	Inhibition of prostaglandin, IL-6, IL-8 and NF-κB signaling	[163,164,165]
Wistar rats, C57BL/6J mice, ex vivo mouse choroidal sprouting model,	Inhibition of angiogenesis	[166,167]

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
