# Peer review of "Vitamin D and Hypoxia: Points of Interplay in Cancer"

_cancers, 2022, doi:10.3390/cancers14071791_

Round 1
Reviewer 1 Report
This is an extensive review regarding vitamin D (specifically calcitriol) and hypoxia in cancer. However, while authors implied the interplay between vitamin D and hypoxia, this review mostly focuses on the interaction between calcitriol and biological pathways that might possibly regulate HIF expression or activation. While authors had done significant amount of works regarding the effect of calcitriol and HIF, it might be better if they use more space to narrate details of their studies in hypoxia, HIF, and how calcitriol plays a role in regulating these pathways more directly. Also, the effect of low oxygen (hypoxia) (not just HIF pathways) on vitamin D, calcitriol, and VDR needs to be included to correlate their actions in tumour microenvironments.
Specific examples of cancer patient data showing the correlation of vitamin D intake or serum vitamin levels vs. survival or treatment outcome would be also informative to expand current knowledge into the clinical applications.
Author Response
Reviewer 1
This is an extensive review regarding vitamin D (specifically calcitriol) and hypoxia in cancer. However, while authors implied the interplay between vitamin D and hypoxia, this review mostly focuses on the interaction between calcitriol and biological pathways that might possibly regulate HIF expression or activation. While authors had done significant amount of works regarding the effect of calcitriol and HIF, it might be better if they use more space to narrate details of their studies in hypoxia, HIF, and how calcitriol plays a role in regulating these pathways more directly. Also, the effect of low oxygen (hypoxia) (not just HIF pathways) on vitamin D, calcitriol, and VDR needs to be included to correlate their actions in tumour microenvironments.
We thank the reviewer for reading and considering our review. Indeed we, initially, aimed to stress out the effects of calcitriol on HIF-mediated signaling since most studies do not take under consideration the effect of hypoxia in their studies (mainly elicited by HIFs). To point-out our intentions we slightely modified the abstract (…..interplay in cellular signaling to give the opportunity…..; line 25).
Furthermore, according to reviewer’s criticism we significantly expanded section 5 (Cancer – Hypoxia – Vitamin D) of the manuscript and included all his suggestions (lines 437-442, 544-546, and 551-588).
Specific examples of cancer patient data showing the correlation of vitamin D intake or serum vitamin levels vs. survival or treatment outcome would be also informative to expand current knowledge into the clinical applications.
We have now added the relevant information according to reviewer’s suggestion (lines 568-579).
Reviewer 2 Report
I read the manuscript under review with interest. The authors raised a very interesting topic relating in general to the interaction between vitamin D and hypoxia. Moreover, the text of the work was enriched with good illustrations.
I only have minor comments:
- The authors should review the work for typos, etc. eg page 3 line 113 - it is 1.24.25 (OH) 4, and it should be (OH)3; p. 9 line 331 - twice “inducing”, line 356: "FIH" instead of "HIF". Etc.
- Authors should standardize the spelling, eg they use HIF-2α or HIF2-α.
- In the chapter "non-genomic response to calcitriol" the authors should review the more recent literature, where, inter alia, the MARRS receptor was found to be PDIA3. There are also studies showing that VDR can locate itself in the caveolae of the cell membrane.
- I also suggest that you extend the legends to figures to be self-readable.
Author Response
Reviewer 2
I read the manuscript under review with interest. The authors raised a very interesting topic relating in general to the interaction between vitamin D and hypoxia. Moreover, the text of the work was enriched with good illustrations.
I only have minor comments:
We thank the reviewer for carefully reading and appreciating our review.
Response to specific comments:
1. The authors should review the work for typos, etc. eg page 3 line 113 - it is 1.24.25 (OH) 4, and it should be (OH)3; p. 9 line 331 - twice “inducing”, line 356: "FIH" instead of "HIF". Etc.
We thank the reviewer. We scanned the document for common errors and corrected them (highlighted text). Where it was needed abbreviations were omitted e.g. FIH is correct and stands for Factor inhibiting HIF.
2. Authors should standardize the spelling, eg they use HIF-2α or HIF2-α.
We thank the reviewer. We scanned the document for and tried to keep universal formatting (highlighted text).
3. In the chapter "non-genomic response to calcitriol" the authors should review the more recent literature, where, inter alia, the MARRS receptor was found to be PDIA3. There are also studies showing that VDR can locate itself in the caveolae of the cell membrane.
We thank the reviewer for his/her useful suggestions. We added additional information and corresponding bibliography to the non-genomic response section (text in highlight; lines 182-185; 189-196).
4. I also suggest that you extend the legends to figures to be self-readable.
We thank the reviewer for the suggestion, we now added descriptive text bellow each legend (highlighted text).
Reviewer 3 Report
The review article titled “Vitamin D and hypoxia: points of interplay in cancer” has discussed the interplay between vitamin D and hypoxia and regulation of HIF-a. The description is comprehensive on this aspect but the association of this axis with cancer is lacking and only briefly discussed. The relation with cancer has been discussed at the end of the review and various pathways discussed in first few sections has not been related to carcinogenesis/tumorigenesis. Further, at various places, the authors have mentioned “cancer cells/lines/animal models” but have not mentioned which cell line and of what origin/which model. Please include a table to show the anticancer effect of vitamin D in-vitro and in-vivo including the model in use (cell line/animal), effect of calcitriol, and the underlying signaling pathway being modulated/mechanism altered.
There are multiple errors in the English language (typos, incomplete sentences, improper word (deregulated is not the proper word to use, the literal meaning of deregulated (remove regulations or restrictions from) doesn’t fits here. It should be dysregulated), vitamin D has been mentioned as Vitamin D multiple times in between sentence (vitamin is not a proper noun), toxic effect of vitamin D-it should be side effect, all abbreviation in between sentences have been expanded with capital letters- e.g., “of Nitric oxide Synthase (iNOS)”- should be inducible nitric oxide synthetase, of several anti‐apoptotic genes such as Bcl2- the authors have mentioned only one gene-please write either Bcl family or mention few others, of apoptotic genes such as Bax- same issue,) Please check carefully the entire manuscript. Few highlights are there as an example.
Author Response
Reviewer 3
The review article titled “Vitamin D and hypoxia: points of interplay in cancer” has discussed the interplay between vitamin D and hypoxia and regulation of HIF-a. The description is comprehensive on this aspect but the association of this axis with cancer is lacking and only briefly discussed. The relation with cancer has been discussed at the end of the review and various pathways discussed in first few sections has not been related to carcinogenesis/tumorigenesis. Further, at various places, the authors have mentioned “cancer cells/lines/animal models” but have not mentioned which cell line and of what origin/which model. Please include a table to show the anticancer effect of vitamin D in-vitro and in-vivo including the model in use (cell line/animal), effect of calcitriol, and the underlying signaling pathway being modulated/mechanism altered.
We thank the reviewer for reading and considering our review. Indeed we, initially, aimed to stress out the effects of calcitriol on HIF-mediated signaling since most studies do not take under consideration the effect of hypoxia in their studies (mainly elicited by HIFs). To point-out our original intentions we slightly modified the abstract (…..interplay in cellular signaling to give the opportunity…..; line 25).
Additionally, according to reviewer’s criticism we significantly expanded section 5 (Cancer – Hypoxia – Vitamin D) of the manuscript according to reviewer’s suggestions (lines 437-442, 544-546, and 551-588 and included the requested table summarizing the effect of calcitriol on various cell lines or animal models. (Table 1; added in line 531).
There are multiple errors in the English language (typos, incomplete sentences, improper word (deregulated is not the proper word to use, the literal meaning of deregulated (remove regulations or restrictions from) doesn’t fits here. It should be dysregulated), vitamin D has been mentioned as Vitamin D multiple times in between sentence (vitamin is not a proper noun), toxic effect of vitamin D-it should be side effect, all abbreviation in between sentences have been expanded with capital letters- e.g., “of Nitric oxide Synthase (iNOS)”- should be inducible nitric oxide synthetase, of several anti‐apoptotic genes such as Bcl2- the authors have mentioned only one gene-please write either Bcl family or mention few others, of apoptotic genes such as Bax- same issue,) Please check carefully the entire manuscript. Few highlights are there as an example.
We corrected the relevant sections of the manuscript (highlighted text).
Round 2
Reviewer 1 Report
Authors included comprehensive information about the possible role of Vitamin D (calcitriol) in treating tumours specifically by focusing on its effect on HIF-related pathways. While this review covers tremendous amount of knowledge regarding the current understanding and future direction of applying calcitriol as a possible treatment for hypoxic or HIF activating cancer, it might be necessary to mention more from the point of renal cancer where HIF pathways are significantly activated. Considering that current clinical trials using HIF2 inhibitors show promising results, as authors mentioned, combination treatment of calcitriol and HIF inhibitors might open a new direction towards these patients.
Author Response
We thank the reviewer for appreciating our review.
In this version, we included minor corrections, and according to the reviewer's suggestion, we expanded section 5. Cancer – Hypoxia – Vitamin D by adding information on ccRCC, HIF inhibitors, clinical trials, and possible interplay with vitamin D levels (lines 597-607; Ref 221-225).
Reviewer 3 Report
The attached manuscript is the same manuscript (v1), please provide the revised manuscript.

Author Response
We apologize revised manuscript uploaded
Round 3
Reviewer 3 Report
None
Author Response
We thank the reviewer for considering our review.
In this version, we included minor corrections and expanded section 5. Cancer – Hypoxia – Vitamin D (lines 597-607).